# WYZE Rule: Federated Rule Dataset for Rule Recommendation Benchmarking

**Mohammad Mahdi Kamani**
Wyze Labs, Inc.
mkamani@wyze.com

**Yuhang Yao**
Carnegie Mellon University
yuhangya@andrew.cmu.edu

**Hanjia Lyu**
University of Rochester
hlyu5@ur.rochester.edu

**Zhongwei Cheng**
Wyze Labs, Inc.
zcheng@wyze.com

**Lin Chen**
Wyze Labs, Inc.
lchen@wyze.com

**Liangju Li**
Wyze Labs, Inc.
lli@wyze.com

**Carlee Joe-Wong**
Carnegie Mellon University
cjoewong@andrew.cmu.edu

**Jiebo Luo**
University of Rochester
jluo@cs.rochester.edu

## Abstract

In the rapidly evolving landscape of smart home automation, the potential of IoT devices is vast. In this realm, rules are the main tool utilized for this automation, which are predefined conditions or triggers that establish connections between devices, enabling seamless automation of specific processes. However, one significant challenge researchers face is the lack of comprehensive datasets to explore and advance the field of smart home rule recommendations. These datasets are essential for developing and evaluating intelligent algorithms that can effectively recommend rules for automating processes while preserving the privacy of the users, as it involves personal information about users' daily lives. To bridge this gap, we present the Wyze Rule Dataset, a large-scale dataset designed specifically for smart home rule recommendation research. Wyze Rule encompasses over 1 million rules gathered from a diverse user base of 300,000 individuals from Wyze Labs, offering an extensive and varied collection of real-world data. With a focus on federated learning, our dataset is tailored to address the unique challenges of a cross-device federated learning setting in the recommendation domain, featuring a large-scale number of clients with widely heterogeneous data. To establish a benchmark for comparison and evaluation, we have meticulously implemented multiple baselines in both centralized and federated settings. Researchers can leverage these baselines to gauge the performance and effectiveness of their rule recommendation systems, driving advancements in the domain. The Wyze Rule Dataset is publicly accessible through HuggingFace's dataset API.

## 1 Introduction

Given the rapid growth of smart devices and applications, there is an increasing need to automate the functionalities of various IoT devices and applications by establishing connections between them. The true value lies in their ability to perform a wide range of tasks and functionalities, tailored to the specific needs and preferences of different users. Configuring device automation to achieve personalized and efficient outcomes becomes a complex endeavor, requiring the assistance of AI and ML techniques. To enable seamless automation, rules serve a critical role. These rules act as predefined conditions or triggers that establish connections between IoT devices, allowing for

37th Conference on Neural Information Processing Systems (NeurIPS 2023) Track on Datasets and Benchmarks.

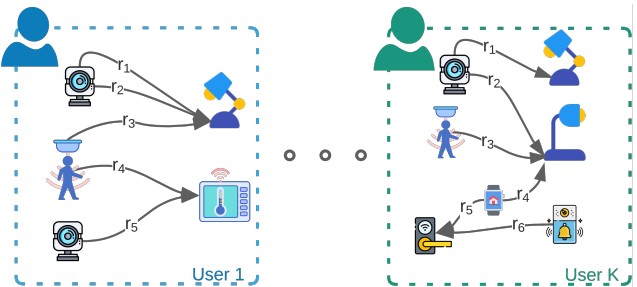

Figure 1: Wyze Rule Dataset - A large-scale dataset for smart home rule recommendation research. The dataset includes over 1 million rules generated by 300,000 users from Wyze Labs, offering a diverse and extensive collection of real-world data. The dataset facilitates the development and evaluation of personalized rule recommendation algorithms for automating smart home processes while respecting user privacy.

customized and automated processes. For example, a rule might dictate that when a motion sensor detects movement, the lights in the room should automatically turn on.

Recommendation systems play a vital role in simplifying, enhancing efficiency, and personalizing the automation process. By recommending rules to users based on their historical interactions with devices and the rules selected by others with similar preferences, the automation process becomes easier, more efficient, and tailored to the specific needs of each user. It is important to note that rule recommendation differs from classical recommendation systems, which primarily rely on a user-item matrix completion approach. In the realm of rule recommendation, four main components are involved: trigger device, trigger state, action device, and action. The main structure of this dataset is shown in Figure 1, where each trigger-action pair is represented as a rule denoted by $r_i$. The triggers and actions can vary depending on the nodes being connected, making the recommendation process more challenging. Specifically, for each pair of trigger and action devices, we aim to recommend an appropriate trigger state and action combination. However, in user-item matrix completion approaches, the complete rule consisting of all four components is treated as a single item. This approach is not able to distinguish between multiple instances of an item within a household, such as different cameras, leading to the recommendation of generic rules.

The mentioned limitations highlight the need for an efficient and personalized rule recommendation system that can consider the unique characteristics of the rule recommendation task. In addition to this, privacy becomes a critical concern as rule recommendation systems require access to user data to provide tailored recommendations. By developing an effective, personalized, and private rule recommendation system, the automation process can be accelerated, leading to higher adoption rates and improved user experience. It is worth mentioning that rule recommendation extends beyond IoT devices and finds applications in diverse automation processes, including software development, workflow automation, task management, smart assistants, and healthcare routine automation.

One of the major challenges faced by researchers in the field of smart home rule recommendations is the lack of comprehensive and scalable datasets. The closest existing dataset available for rule recommendation research is the IFTTT dataset [14]. This dataset primarily focuses on rules connecting various apps, such as Gmail and Instagram, for users. It consists of rules created by a subset of users and adopted by a larger user base within the IFTTT platform. The IFTTT dataset was initially extracted from the platform in September 2015 by Ur et al. [14] and subsequently explored by Corno et al. [3]. It comprises a total of 151 users, 156 devices, 500 triggers, 235 actions, and 4,015 known rules. Although the IFTTT dataset has garnered attention from researchers in the recommendation systems domain, its small scale limits its practicality for real-world applications. To overcome these limitations and bridge the gap between research and practical applications, there is a pressing need for a more extensive and representative dataset, which the Wyze Rule Dataset aims to fulfill.

Introducing the Wyze Rule Dataset, we present a large-scale and user-centric dataset specifically designed for smart home rule recommendation research. The Wyze Rule Dataset comprises an extensive collection of over 1 million rules, created by individual users to meet their unique needs within their smart homes. This dataset is sourced from Wyze Labs and encompasses a diverse user base of 300,000 individuals, providing a rich and varied pool of real-world data. Unlike the IFTTT

dataset, the Wyze Rule Dataset reflects the specific device configurations of each user, including the possibility of multiple instances for each device type. Each user's set of available and valid rules can differ significantly from others due to the distinct combinations of devices. This inherent variability adds a layer of complexity to rule recommendation, as each pair of devices may have its own set of valid rules that differs from those of other device pairs. For example, a motion sensor and a camera may have a valid trigger-action pair of "motion detected-turn on," which may not be applicable to a contact sensor and a thermostat.

The individualized nature of the rules generated by users, based on their unique device setups, presents an excellent opportunity to explore federated learning approaches [10, 6, 4, 5]. Federated learning enables the development of personalized recommendation models without the need to transfer rules and usage data to an external server. With the wide range of device combinations within each user's household and the resulting diversity of valid rules, the distribution of data among users in the Wyze Rule Dataset exhibits a high level of heterogeneity. Consequently, the dataset offers a compelling real-world use case for developing federated learning algorithms that effectively address the challenges posed by data heterogeneity among users.

The Wyze Rule Dataset is made available under the "CC BY-NC-ND" license, specifically for non-commercial and research-related purposes. To obtain access to the dataset, interested individuals are required to complete and sign the consent form, which can be accessed through the following link: Wyze Rule Recommendation Dataset. Once the consent form is submitted, the dataset will be released to the approved individuals, enabling them to utilize it for their research endeavors.

## 2   Related Work

The majority of recommendation systems primarily rely on collaborative filtering techniques, such as matrix factorization approaches [9, 2], to match users with items of interest. These systems consider user and item features, as well as historical data reflecting user preferences for a given set of items. Recently, the adoption of graph neural networks has become prevalent in recommendation systems, particularly for modeling relationships using bipartite graph structures. One notable graph-based auto-encoder framework for matrix completion problems is GCMC [1]. It utilizes graph structures to provide recommendations. R-GCNs [12], on the other hand, leverage graph convolutional networks to model relational data by considering different relations between users and items. For dealing with heterogeneous graphs containing user and item features, HetGNN [19] has been proposed. Additionally, RotatE [13] introduces a seminal approach for knowledge graph embeddings by employing relational rotation in a complex number space.

With the advancements in smart home technology, rule recommendation systems for IoT devices have gained significant practical and research potential [18, 17]. IFTTT, a mobile app offering rule execution services, provides an open dataset of users and rules for analysis [14, 11]. However, this dataset primarily consists of popular rules created by some users and utilized by others, with fixed entities. In contrast, the Wyze Rule dataset consists of rules created by individual users to cater to their specific needs. The entities in this dataset can have multiple instances, such as a user having multiple different cameras. Furthermore, the scale of the Wyze Rule dataset is much larger than that of the IFTTT dataset. For rule recommendation in IFTTT, RecRule [3] presents a recommendation approach based on a user-rule matrix and employs semantic reasoning to enrich the rules with additional semantic information. TaKG [16], on the other hand, constructs a trigger-action knowledge graph and utilizes collaborative filtering between users and rules to recommend appropriate rules. GraphRule and FedRule, recently introduced in Yao et al. [17], introduce a user-centric graph structure where each user has their own graph. In this setting, the task of rule recommendation transforms into a link prediction problem.

## 3   Wyze Rule Dataset

### 3.1   Meta Data

The dataset consists of two files, `rule.csv`, and `device.csv`, which provide information about the rules that control the behavior of Wyze smart home devices and the devices owned by users, respectively. The structure of each of these two files is as follows:

- **Rule Dataset**: This file (`rule.csv`) contains data related to the rules that govern the behavior of Wyze smart home devices. Each row represents a single rule and contains various attributes describing the rule. The attributes of this file are as follows:
    - '`user_id`' (`int`): A unique integer identifier for the user associated with the rule. This identifier has been anonymized and does not contain any information related to the Wyze users.
    - '`trigger_device`' (`str`): The model of the device that triggers the rule when a specific condition is met. It may be a Wyze smart home device such as a sensor or a camera.
    - '`trigger_device_id`' (`int`): A unique integer identifier for the trigger device.
    - '`trigger_state`' (`str`): The state or condition that needs to be met on the trigger device for the rule to be activated. It may represent values such as "on," "off," "motion detected," or "sensor open."
    - '`action_device`' (`str`): The model of the device that performs an action when the rule is triggered. It is a Wyze smart home device such as a light or a camera.
    - '`action_device_id`' (`int`): A unique integer identifier for the action device.
    - '`action`' (`str`): The action to be executed on the action device when the rule is triggered. It may include values like "power on," "power off," "start recording," or "change brightness."
- **Device Dataset**: This file contains data related to the devices owned by users. Each row represents a single device and contains information about the device model and its association with a specific user. There are a number of devices in this dataset that are not used in any rules by users, and hence, are not present in the rule dataset. The attributes of this dataset are as follows:
    - '`user_id`' (`int`): A unique integer identifier for the user associated with the device.
    - '`device_id`' (`int`): A unique integer identifier for the device.
    - '`device_model`' (`str`): The model or type of the device owned by the user. It represents various Wyze smart home devices such as a camera, a sensor, or a switch.

## 3.2 Dataset Statistics

This dataset comprises the rules that establish connections between smart devices in various clients' households. As a result, the distribution of rules among clients is non-IID, making it suitable for federated learning settings. The dataset, referred to as the "Wyze Rule" dataset, encompasses data from over $300,000$ users with more than $1$ million rules. To address privacy concerns, we have thoroughly cleansed the dataset, removing all personally identifiable information (PII).

The rules in the dataset have been simplified and follow a specific structure: "$<$ trigger entity, trigger-action pairs, action entity $>$". The trigger-action pairs represent the type of connection established between devices. For instance, an example rule could be connecting a smart doorbell to a camera, where pressing the doorbell triggers the camera to power on for recording. In this case, the rule format would be "<Doorbell, Doorbell Pressed - Power On, Camera>". The dataset encompasses 16 distinct device types and $1,323$ unique trigger-action pairs, resulting in $2,968$ unique rules. Table 1 provides a detailed breakdown of the entity types and samples of trigger-action pairs. It is worth noting that the entity "Cloud" represents an arbitrary device where the action involves modifying settings on the same device in the app. For instance, in the rule "When the Camera detects motion, turn on the notification," the action device would be classified as "Cloud" since it changes the setting of that device in the app.

As it was mentioned, the data in this dataset is heavily non-IID among users. As evidence, in Figure 2 the distribution of the number of rules and devices among users can be seen. The distribution is a long-tailed one, indicating, the heterogeneity of data distribution among users, making this real-world dataset a perfect use case for federated learning algorithms. In the dataset, there are some rare users, with more than 200 rules, which for the sake of visualization, we did not include in the distribution in Figure 2a. Note that the Cloud only appears in the action device list.

The data heterogeneity can also be seen from the distribution of device types owned by users, as well as utilized by rules set by users. The share of device types owned by users can be seen in Figure 3a, in

| Types of Entities | Types of Trigger-Action Pairs |
|---|---|
| Camera | Open, Turn On |
| Climate Sensor | Open, Turn Off |
| Contact Sensor | Open, Turn on motion detection |
| Irrigation | Open, Change Brightness |
| Leak Sensor | ... |
| Light | Detects a person, Unmute notification |
| Light Strip | Detects a pet, Unmute notification |
| Lock | Detects a vehicle, Unmute notification |
| Mesh Light | ... |
| Motion Sensor | Open, Turn on the siren |
| Outdoor Plug | Open, Turn off the siren |
| Plug | ... |
| Robot Vacuum | Detects a person, Turn off |
| Switch | Detects a package, Turn on |
| Thermostat | Detects smoke alarm sound, Turn off |
| Cloud | (Doorbell) Is pressed, Turn On |

Table 1: Wyze Rule Dataset: Device list and trigger-action pairs samples.

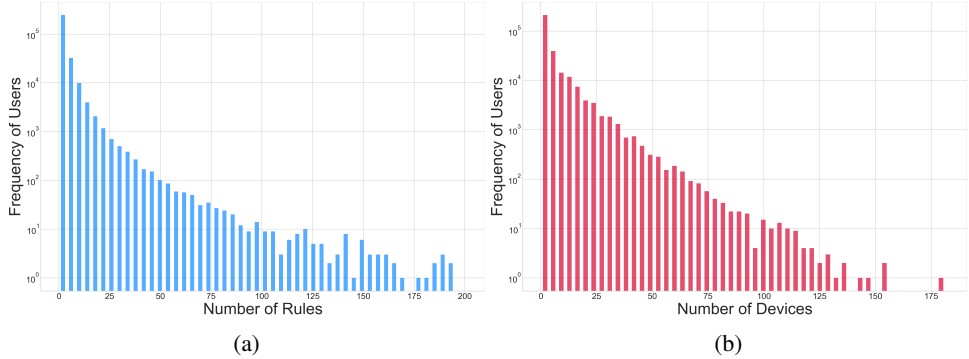

(a)  (b)

Figure 2: The distribution of the number of Rules and Devices among users in Wyze Rule Dataset.

which, Camera is the most owned device among users, followed by Plug and Light. The distribution of device types utilized by rules is fairly similar, visible in Figure 3b. As it is expected, and can be seen from Figures 3c and 3d, after Camera, sensors (contact sensor and motion sensor) are the most used devices as triggers by the users. Light and plug are the most used action devices after Camera and Cloud in the rules.

## 3.3 Metrics

To evaluate models trained on this dataset, similar to other recommendation models, we use various offline metrics. These metrics are mainly designed to evaluate the ability of the model to predict omitted rules from the database with a higher probability. The metrics are defined as follows:

**AUC** The Area Under the ROC (Receiver Operating Characteristic) curve represents the overall performance of the rule recommendation model. It is a widely-used metric in machine learning evaluation. A higher AUC indicates a better ability of the model to rank rules correctly, distinguishing positive and negative recommendations effectively.

**Positive Mean Rank** The positive mean rank is a metric that quantifies the average position or rank of the ground truth rules within the list of recommended rules over all users' data. It specifically focuses on the rank of the positive rules, which are the rules that should ideally be included in the recommendations. A lower positive mean rank indicates better performance, as it signifies that the ground truth rules are ranked closer to the top of the recommendation list. Improving the positive

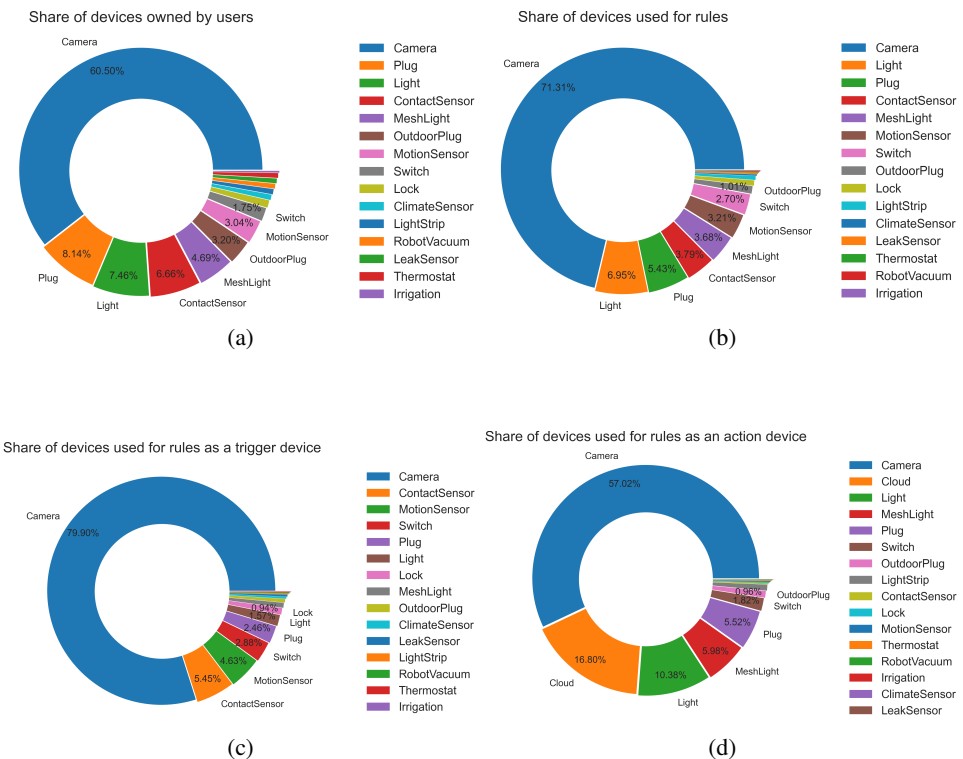

Figure 3: Share of different device types among users and utilized by rules either as a trigger device or action device.

mean rank metric helps enhance the overall user experience by ensuring that the most important and useful rules are presented prominently in the recommendation results.

**Mean Reciprocal Rank**   Mean Reciprocal Rank (MRR) is a metric commonly used to evaluate the performance of a recommendation system. It measures the average of the reciprocal ranks of the ground truth rules within the recommendation list. MRR takes into account the position or rank of the first correctly recommended rule, considering a lower rank as more desirable. We average this measure over all users' data:

$$\text{MRR} = \frac{1}{M} \sum_{m \in [M]} \frac{1}{|\mathcal{T}_m|} \sum_{i \in \mathcal{T}_m} \frac{1}{\text{rank}_i}, \tag{1}$$

where $\mathcal{T}_m$ represents the set of ground truth rules that are omitted from the data of user $m$. By considering the reciprocal rank, MRR encourages recommendation systems to prioritize the most relevant rules and rank them higher in the recommendation list. This aligns more closely with user preferences and improves the overall user experience by increasing the chances of important rules being discovered earlier.

**Weighted Mean Reciprocal Rank**   Weighted Mean Reciprocal Rank (WMRR) is an extension of the Mean Reciprocal Rank (MRR) metric that incorporates the notion of weights assigned to different ground truth rules. In this case, for each two nodes, the number of valid rules is different. Hence, by using this number as a weight in MRR, we can distinguish between the recommendations that have the same rank but over very different size of available valid rules.

$$\text{WMRR} = \frac{1}{M} \sum_{m \in [M]} \frac{1}{\sum_{i \in \mathcal{T}_m} w_i} \sum_{i \in \mathcal{T}_m} \frac{w_i}{\text{rank}_i}, \tag{2}$$

where $w_i$ is the number of valid rules between two nodes of the $i$-th rule in the omitted ground truth rule set $\mathcal{T}_m$.

# 4 Benchmarks

To demonstrate the potential of this dataset, we employ it in various training procedures aimed at real-world scenarios of rule recommendation. We utilize the defined metrics to assess the performance of these trained models on the dataset and subsequently explore the possibilities and opportunities associated with using this dataset for such tasks.

## 4.1 Experiment Setting

We investigate two training modes: centralized learning and federated learning, aiming to demonstrate the dataset's potential for graph representation learning in the federated learning domain. As discussed in this paper and by Yao et al. [17], we have two options: treating each rule as an individual item and applying user-item matrix completion approaches (similar to conventional recommendation systems), or constructing a graph for each user and conducting link prediction tasks. In this study, we mainly focus on graph representation learning, using the same settings as described by Yao et al. [17]. For training, we primarily utilize the Adam optimizer with a learning rate of 0.1 and train for 100 epochs. In the case of federated learning algorithms, we adopt 3 local steps during each communication round, totaling 300 local iterations over 100 communication rounds.

## 4.2 Centralized Benchmarks

We conduct a comparison between three graph representation learning models using the proposed GraphRule [17], a centralized optimization for graph-based link prediction of rule recommendation. It is important to note that due to the different task structures, direct comparison between graph representation learning approaches and user-item matrix completion methods is challenging. In the matrix completion approaches, each rule is treated as an individual item, irrespective of the trigger and action nodes involved. This means that two rules with the same trigger and action, but associated with different sets of motion sensors and cameras within a house, would be considered as one item in matrix completion approaches. However, in graph-based approaches, these rules would be represented as separate edges, distinguishing them based on the specific combinations of sensors and cameras.

| | Hit Rate @50 | Hit Rate @50 (Multiple Devices with Same Type) |
|---|---|---|
| Matrix Factorization | 0.8138 | NA |
| GCMC | 0.8445 | NA |
| GraphRule | 0.91 | 0.6396 |

Table 2: Hit Rate for different baselines. Classical user-item matrix completion approaches cannot distinguish between multiple devices with the same type in rules for a user.

As shown in Table 2, when using user-item completion (Matrix Factorization and GCMC) in the rule recommendation setting, since there is only one user-item relationship (user, select, rule item) while they cannot deal with multiple devices with the same type, which makes the performance of user-item completion method very limited. When recommending 50 rule, the hit rate of Matrix Factorization and GCMC become 0 since it only does not consider multiple devices with the same type.

For the GraphRule [17] approach we use three different models as its GNN representation model. The models are: GraphSAGE [7], Graph Attention Network (GAT) [15], and Graph Convolutional Network (GCN) [8]. For each of these models, we use two layers of neural networks with a hidden layer size of 32. The input size is 16, which is the one-hot-encoding size of the device types. The output size is $1,323$, which is the number of unique trigger-action pairs in the dataset. For the GAT model, we use 5 heads. We run the training for 100 epochs. Figure 4 shows the results of applying the GraphRule approach to the Wyze Rule Dataset. We provide a comparison between different GNN models on this approach on all different metrics we introduced in 3.3. GraphSAGE consistently outperforms the other models across various metrics, indicating its superior performance. It is worth noting that while there is a significant disparity in the MRR metric, weighted MRR offers a more accurate comparison and demonstrates their competitiveness.

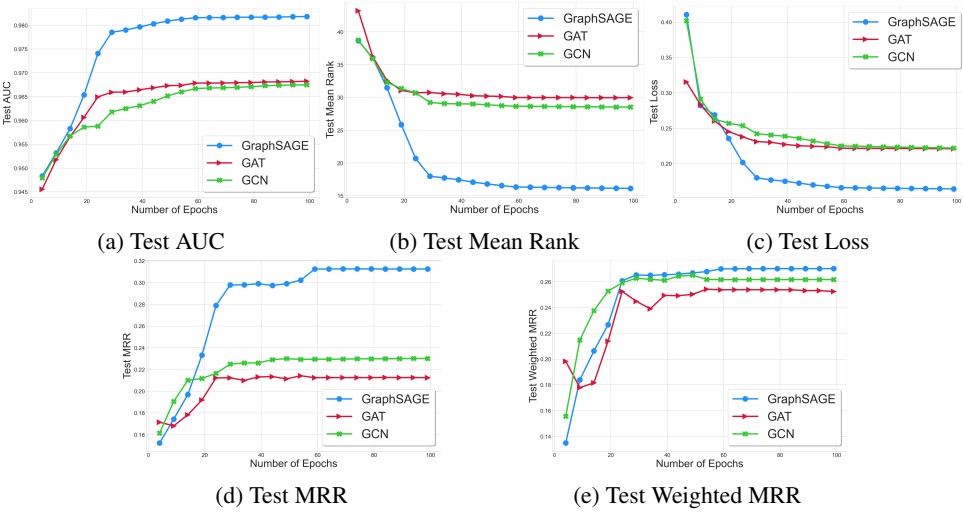

(a) Test AUC       (b) Test Mean Rank       (c) Test Loss

(d) Test MRR       (e) Test Weighted MRR

Figure 4: Test results of applying GraphRule algorithms with different GNN models for node representation on Wyze Rule Dataset.

## 4.3 Federated Benchmarks

To demonstrate the potential of the Wyze Rule Dataset in the realm of federated learning algorithms, we employ the proposed FedRule algorithm introduced by Yao et al. [17]. Building upon the GraphSAGE model utilized in the previous step, FedRule leverages a federated learning framework where each user independently updates the model on their local data and subsequently transmits the updated models to a central server for aggregation. The Wyze Rule Dataset presents a unique challenge for federated learning due to the high heterogeneity in the data distribution and variations in data sizes across different users. This heterogeneity poses a significant obstacle to achieving comparable performance to centralized approaches in federated learning. However, addressing this challenge is crucial as federated learning techniques offer the potential to preserve user privacy while achieving comparable performance to centralized approaches.

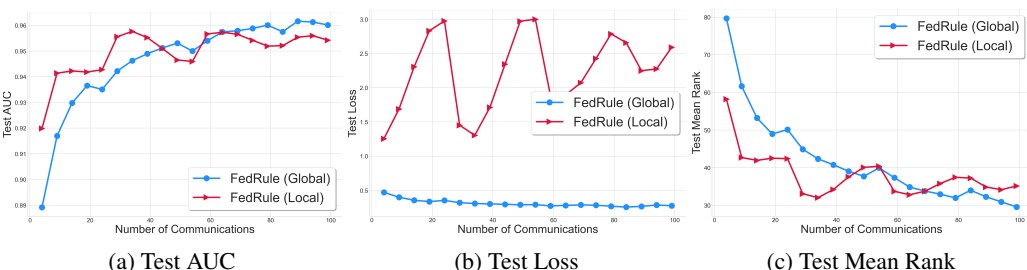

(a) Test AUC       (b) Test Loss       (c) Test Mean Rank

Figure 5: Test results of applying FedRule algorithms with GraphSAGE models for node representation on Wyze Rule Dataset.

Figure 5 illustrates the performance dynamics of the local models and the global model in the federated learning process. Due to the significant heterogeneity among clients' data, the local models exhibit a ripple effect with high variance. In contrast, the global model, which is obtained through the aggregation of the local models, demonstrates a smooth convergence. The comparison of Mean rank values between the FedRule approach and the centralized training provides valuable insights. After 100 rounds of communication, the Mean rank on FedRule is approximately 30, which is nearly twice the value observed in the centralized training. This substantial difference highlights the existence of a noticeable gap between the two modes of training. The observations derived from these results emphasize the importance of developing a specialized federated learning approach that can effectively address the high heterogeneity in the data distribution within the Wyze Rule dataset. Such an approach should aim to mitigate the impact of data heterogeneity and strive to achieve performance levels that are closer to those obtained in centralized training scenarios.

### 4.4 Evaluation against the Prior Version of the Wyze Rule Dataset

The initial introduction of the Wyze Rule dataset took place in the FedRule paper [17], where the novel GraphRule and FedRule algorithms were proposed, outperforming traditional recommendation techniques. While the GraphRule, incorporating a 2-layer GraphSage model paired with a 2-layer Neural Network predictor, exhibits excellent performance on smaller datasets, its application to our newly released, more expansive dataset poses certain challenges. It's noteworthy to mention that, despite these challenges, GraphRule continues to surpass other methodologies on this new dataset. Nevertheless, the overall efficacy of all methods seems to diminish due to the intricate nature of this new large-scale dataset.

**Challenge 1: The Necessity for Enhanced Centralized Rule Recommendation Algorithms**
Applying the existing model to the expansive cross-device dataset results in a noticeable decrease in the Mean Rank, which is concerning given the application of recommendation for this dataset.

|  | Test Loss | AUC | Mean Rank |
|---|---|---|---|
| GraphRule (small dataset) | 0.1997 | 0.9768 | 4.349 |
| GraphRule (large dataset) | 0.1911 | 0.9825 | 10.46 |

Table 3: Performance comparison of the GraphRule model on small and large datasets, showcasing the variation in Test Loss, AUC, and Mean Rank values.

There's a clear need for more refined algorithms, prompting our interest in hosting a challenge[1] centered around this dataset. An additional observation suggests that focusing on methods that optimize the mean rank directly, rather than merely addressing test loss, could prove beneficial.

**Challenge 2: Addressing Cross-device Federated Learning with Scarce Data and High Data Heterogeneity**

In our previous research [17], when employing FedGate as the training method, FedRule achieved performance equivalent to centralized training. However, using FedAvg as the training method caused FedRule not to converge. Upon deploying on the larger dataset, FedRule, regardless of whether it employs FedAvg or FedGate, can converge. Yet, it fails to match the performance of centralized training. This indicates that both FedAvg and FedGate struggle in such cross-device contexts.

|  | Test Loss | AUC | Mean Rank |
|---|---|---|---|
| FedRule (small dataset, with FedAvg) | 0.3878 | 0.9521 | 6.661 |
| FedRule (small dataset, with FedGate) | 0.1892 | 0.9804 | 4.156 |
| FedRule (large dataset, with FedAvg) | 0.2854 | 0.9613 | 30.92 |
| FedRule (large dataset, with FedGate) | 0.2907 | 0.9486 | 37.99 |

Table 4: Performance metrics of the FedRule model using different training methods (FedAvg and FedGate) across small and large datasets, highlighting the differences in Test Loss, AUC, and Mean Rank.

In summary, the larger dataset illuminates the path for crafting superior centralized rule recommendation algorithms, formulating novel Federated Learning algorithms for cross-device scenarios, and jointly enhancing both areas.

## 5 Discussion

The Wyze Rule Dataset presents exciting research opportunities in the field of rule recommendation. By leveraging the rich and diverse set of rules created by individual users, researchers can develop and evaluate novel recommendation algorithms that cater to the specific needs and preferences of smart home users. In addition, the availability of this dataset introduces a novel evaluation opportunity for

---

[1]The Wyze Rule Recommendation Challenge is taking place on the HuggingFace website from August 15, 2023, to November 10, 2023. For details, visit: `https://huggingface.co/spaces/competitions/wyze-rule-recommendation`.

federated learning algorithms, as it inherently exhibits non-IID characteristics. This dataset enables the exploration of advanced techniques for personalized rule recommendations, taking into account the heterogeneous device configurations and unique user contexts present in real-world smart homes. The Wyze Rule Dataset also opens up opportunities for investigating 'node recommendations', which involves suggesting suitable devices or sensors for specific automation tasks. By analyzing the relationships between devices, triggers, and actions in the dataset, researchers can develop algorithms that recommend the most appropriate nodes to users based on their specific requirements. This can help users make informed decisions when expanding their smart home ecosystems and further improve the automation capabilities of their setups.

## 6 Limitations

One limitation of this dataset is that the location of each device is not indicated, hence, we do not know which devices are co-located together (i.e., in the same room or in the same house). This will make the recommendation more difficult since it might recommend valid rules that are not useful. Another limitation worth considering is the absence of direct human feedback or online comparison data within the dataset. While the Wyze Rule Dataset provides valuable information about the rules created by users, it does not include explicit feedback on the effectiveness or satisfaction of the recommended rules. Incorporating user feedback and conducting online comparisons could provide deeper insights into real-world performance and user preferences regarding rule recommendations.

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

## Dataset Papers Checklist

1. Submission introducing new datasets must include the following in the supplementary materials:

   (a) Dataset documentation and intended uses. Recommended documentation frameworks include datasheets for datasets, dataset nutrition labels, data statements for NLP, and accountability frameworks. A: Available on the dataset page.

   (b) URL to website/platform where the dataset/benchmark can be viewed and downloaded by the reviewers. A: The dataset is hosted on HuggingFace dataset API: Wyze Rule Recommendation Dataset

   (c) Author statement that they bear all responsibility in case of violation of rights, etc., and confirmation of the data license. A: The consent form in the dataset webpage indicates its intended use and terms of access to the dataset.

   (d) Hosting, licensing, and maintenance plan. The choice of hosting platform is yours, as long as you ensure access to the data (possibly through a curated interface) and will provide the necessary maintenance. A: The dataset will be hosted on the HuggingFace repository and will be maintained and updated from there.

2. To ensure accessibility, the supplementary materials for datasets must include the following:

   (a) Links to access the dataset and its metadata. This can be hidden upon submission if the dataset is not yet publicly available but must be added in the camera-ready version. In select cases, e.g when the data can only be released at a later date, this can be added afterward. Simulation environments should link to (open source) code repositories. A: All are available on the Wyze Rule Recommendation Dataset repository on HuggingFace.

   (b) The dataset itself should ideally use an open and widely used data format. Provide a detailed explanation of how the dataset can be read. For simulation environments, use existing frameworks or explain how they can be used. A: The data files are stored in a CSV format. They are easily accessible via HugginFace's dataset Python library.

   (c) Long-term preservation: It must be clear that the dataset will be available for a long time, either by uploading to a data repository or by explaining how the authors themselves will ensure this. A: The dataset will be available on the HuggingFace website.

(d) Explicit license: Authors must choose a license, ideally a CC license for datasets, or an open-source license for code (e.g. RL environments). **A: This dataset is licensed by cc-by-nc-nd-4.0, which prohibits commercial use, distribution, modification, and reproduction of the data without permission from the copyright holder.**

(e) Add structured metadata to a dataset's meta-data page using Web standards (like schema.org and DCAT): This allows it to be discovered and organized by anyone. If you use an existing data repository, this is often done automatically. **A: The meta data is available on the repository page.**

(f) Highly recommended: a persistent dereferenceable identifier (e.g. a DOI minted by a data repository or a prefix on identifiers.org) for datasets, or a code repository (e.g. GitHub, GitLab,...) for code. If this is not possible or useful, please explain why. **Since it can be uniquely identified by the HuggingFace dataset API it does not seem necessary at the moment.**

3. For benchmarks, the supplementary materials must ensure that all results are easily reproducible. Where possible, use a reproducibility framework such as the ML reproducibility checklist, or otherwise guarantee that all results can be easily reproduced, i.e. all necessary datasets, code, and evaluation procedures must be accessible and documented. **The code for the benchmarks will be available soon through the repository website.**

4. For papers introducing best practices in creating or curating datasets and benchmarks, the above supplementary materials are not required.

