# OpenReview forum: "Wyze Rule: Federated Rule Dataset for Rule Recommendation Benchmarking"
_NeurIPS.cc/2023/Track/Datasets_and_Benchmarks — NeurIPS 2023 Datasets and Benchmarks Poster_

### Official Review · Reviewer_e1DN · 2023-06-30

**Rating:** 7
**Confidence:** 3
**Clarity:** The paper is overall well-written and…

**Strengths:**

- The paper is well-written and easy to read. In particular, the paper reasonably motivates the need of a large-scale federated rule recommendation dataset, and clearly highlights the advantages of Wyze Rule Dataset in comparison to prior work, notably IFTTT dataset.

- The instructions to get access to the data are clear and easy-to-follow. Moreover, the terms of use **seem** to protect against the misuse of the dataset and would incentivize a fair usage. (The use of the verb "seem" is intentional, as I am not a legal expert)

- The dataset has a natural federated split among clients and showcases a certain level of data heterogeneity, which makes it suitable for federated learning benchmarking. The dataset has the potential to become a standard benchmark for federated rule recommendation.

- The paper discusses the limitation of Wyze Rule Dataset and provide a reasonable plan to mitigate these limitations.

**Additional Feedback:**

I thank the authors for the effort, and encourage them to incorporate some (or all) of the suggestions discussed above in order to improve the paper/benchmark.

**Correctness:**

The claims of the paper are in general correct. The only exception is the claim that "the distribution [shown in Figure 2] is a long-tailed one, indicating the heterogeneity of data distributions among users". I disagree with this claim, because the Figure 2 shows the distribution of the clients' local datasets sizes. In fact, even if clients have different number of samples, these samples might be identically distributed.

On a different note, I think that the code might have some errors, for example the variable `fedtype` in `training.py/local_fedAvg` is not defined.

**Documentation:**

The documentation is clear and easy to follow. The guidelines to obtain data are precise.

**Ethics:**

I discuss potential privacy leakage above.

**Limitations:**

Besides the limitations discussed in Section 6, one can add the privacy concerns resulting from the public release of Wyze Rule Dataset. While I am not an expert on data anonymization techniques, I believe that removing users identifiers from the dataset is not always a privacy/anonymization guarantee as information regarding the users could be retrieved by  combining multiple public datasets; see for example [2].

[2] Sweeney, Latanya. "Simple demographics often identify people uniquely." Health (San Francisco) 671.2000 (2000): 1-34.

**Opportunities For Improvement:**

Besides the limitations discussed in Section 6, the paper/benchmark could be improved:
1) the paper could quantify the statistical heterogeneity of the dataset. Prior federated benchmarks, e.g., FLamby [1], provides tools to quantify the statistical heterogeneity in federated benchmarks.
2) the code-base could be improved by following PEP coding conventions for Python (e.g., number of empty lines, functions names..), and by removing dead pieces of the code.
3) the paper could explore the hyper-parameters space.


[1] du Terrail, Jean Ogier, et al. "FLamby: Datasets and Benchmarks for Cross-Silo Federated Learning in Realistic Healthcare Settings." Thirty-sixth Conference on Neural Information Processing Systems Datasets and Benchmarks Track.

**Relation To Prior Work:**

The paper discusses enough related work, and highlights the differences with other rule recommendation benchmarks.

**Summary And Contributions:**

The paper introduces Wyze Rule Dataset; a federate dataset for smart home rule recommendation. The dataset encompasses 1 million rules (i.e., triplets of trigger entity, trigger-action pair and action entity representing connections between IoT devices present in smart homes) gathered from 300,000 individuals/clients.

The advantage of Wyze Rule Dataset in comparison to prior rule recommendation datasets, such as IFTTT, is two folds:
1) Wyze Rule Dataset is a large-scale dataset with 1 million rules. In comparison, IFTTT comprises of 4,015 rules.
2) Wyze Rule Dataset has a natural federated split among 300,000 clients; therefore can be considered as a benchmark for federated rule recommendation algorithms.

The paper provides a few statics of Wyze Rule Dataset to demonstrate the natural statistical heterogeneity present in the dataset. Additionally, the paper runs a few numerical experiments to 1) compare the performance of three graph representation learning models in the centralized case, and 2) demonstrate that the statistical heterogeneity of Wyze Rule Dataset poses a challenge forr federated training.

---

> ### Author Response · Authors · 2023-08-21
>
> Thank you for the valuable comments and suggestions!
>
> >**The paper could quantify the statistical heterogeneity of the dataset. Prior federated benchmarks, e.g., FLamby [1] provide tools to quantify the statistical heterogeneity in federated benchmarks.**
>
> Great suggestion. We will add such analysis of the heterogeneity of our dataset to the paper.
>
> >**The code-base could be improved by following PEP coding conventions for Python (e.g., number of empty lines, functions names..), and by removing dead pieces of the code.**
>
> Thanks for pointing that out. This is not the final version of our code. We already have released the dataset for public access here (https://huggingface.co/datasets/wyzelabs/RuleRecommendation). We will release the cleaned version of our code soon.
>
> >**The paper could explore the hyper-parameters space.**
>
> Thanks for mentioning this. Since we are providing a benchmark for other researchers to use and develop their own models and algorithms on top of this dataset, we are not fully optimizing for all the hyperparameters involved. The main goal of this benchmark paper is the dataset itself, and not the learning algorithm. For the algorithm focus on a part of this dataset we have published another paper here: https://dl.acm.org/doi/abs/10.1145/3576842.3582328
>
> >**Besides the limitations discussed in Section 6, one can add the privacy concerns resulting from the public release of Wyze Rule Dataset. While I am not an expert on data anonymization techniques, I believe that removing users identifiers from the dataset is not always a privacy/anonymization guarantee as information regarding the users could be retrieved by combining multiple public datasets; see for example [2].**
>
> That is a valid concern. The privacy of our users is our first priority. The case of privacy leaks such as one similar to the Netflix recommendation case might not happen in our dataset, since we have removed any user-related information such as their id, device names, their location, etc. Hence, we can safely use this dataset for research purposes.
>
> >**The claims of the paper are in general correct. The only exception is the claim that "the distribution [shown in Figure 2] is a long-tailed one, indicating the heterogeneity of data distributions among users". I disagree with this claim, because Figure 2 shows the distribution of the clients' local datasets sizes. In fact, even if clients have different number of samples, these samples might be identically distributed.**
>
> Thanks for pointing this out. In fact, we are referring to the distribution of local clients' data sizes. For showing the statistical heterogeneity, we use approaches like gradient similarity or the one you mentioned.
>
> >**On a different note, I think that the code might have some errors, for example the variable fedtype in training.py/local_fedAvg is not defined.**
>
> As we mentioned, the final version of the code will be released soon

---

### Official Review · Reviewer_qebj · 2023-07-20

**Rating:** 5
**Confidence:** 4
**Correctness:** This paper seems correct sufficiently.

**Strengths:**

They open a large dataset that can be useful for smart home settings.

**Additional Feedback:**

See other comments.

**Clarity:**

It is hard to understand "rules" in the introduction. Also, it is unsure why related work starts describing recommendation systems. The main contribution should be Wyze rules dataset, so the authors first should explain datasets related to Wyze rule dataset.

**Documentation:**

Their appendix includes a simple ReadMe file.

**Ethics:**

No problem.

**Limitations:**

The Wyze rule datasets do not include time information. Time is important to infer rules in smart home applications.

**Opportunities For Improvement:**

1. It is hard to understand rules from the introduction. For example, Figure 1 seems a just graph between devices.
2. The authors mentioned R-GCN, GCMC, RotatE, and HetGNN in related works, but in benchmarking experiments, different methods are used as baselines. The authors should use these methods in benchmarking.
3. It is better to discuss what we can and cannot do with and without Wyze rules dataset, respectively. Also, it is better to evaluate the performance difference of existing methods between Wyze rule and IFTTT datasets, in terms of scalability, efficiency, and accuracy.

**Relation To Prior Work:**

Yes, it is sufficiently discussed about other datasets.

**Summary And Contributions:**

This paper introduces a new dataset for smart home actions from users to devices, called Wyze rule dataset. This dataset is quite larger compared with existing ones. Also, the authors evaluate the performance of graph neural networks in centralized and federated settings by using Wyze datasets.

---

> ### Author Response · Authors · 2023-08-21
>
> Thank you for the comments and valuable suggestions!
>
> >**It is hard to understand the rules from the introduction. For example, Figure 1 seems a just graph between devices.**
>
> Thanks for pointing that out. We will add more clarifying introduction about the rules, i. e. , what they consist of, and how they are used in practice.
>
> >**The authors mentioned R-GCN, GCMC, RotatE, and HetGNN in related works, but in benchmarking experiments, different methods are used as baselines. The authors should use these methods in benchmarking.**
>
> The main purpose of the paper is to benchmark centralized and federated training, which can use any ML model like GCN, GraphSage, GAT, R-GCN, and GCMC. We do have a comparison of different methods in our other paper (https://dl.acm.org/doi/abs/10.1145/3576842.3582328), where an in-detail comparison between different algorithms on this dataset can be found. We are hoping to attract more researchers to develop new algorithmic solutions to tackle this recommendation task and improve over our benchmarks. That is why we are hosting a competition for this task (https://huggingface.co/spaces/competitions/wyze-rule-recommendation)
>
> >**It is better to discuss what we can and cannot do with and without Wyze rules dataset, respectively. Also, it is better to evaluate the performance difference of existing methods between Wyze rule and IFTTT datasets, in terms of scalability, efficiency, and accuracy.**
>
> More discussions sound very helpful for the paper. We will add analysis on the heterogeneity of data distribution. Detailed dataset information is in the dataset description (https://huggingface.co/datasets/wyzelabs/RuleRecommendation). Comparing existing methods on Wyze rule and IFTTT datasets are also in our other paper (https://dl.acm.org/doi/abs/10.1145/3576842.3582328).
>
> >**The Wyze rule datasets do not include time information. Time is important to infer rules in smart home applications.**
>
> Understanding the temporal information related to rules is a valid point, also mentioned by reviewer YUfZ. This can show how the rules are changed over time. However, as discussed before in the case of our smart home rules, the rules are not changing frequently, and the dataset gathered for this benchmark consists of fairly stable rules that have been used by the users for a long time.

---

> > ### Comment · Reviewer_qebj · 2023-08-22
> >
> > Thanks for your sincere reply.
> > After reading your reply, I am confused about your contributions. I supposed that the main contribution is building and opening a new dataset, i.e., Wyze rule dataset. However, it seems that the paper related to Wyze rule dataset is already published in other conferences. In addition, benchmarking also seems done in other papers. Thus, the Wyze rule dataset and benchmarking do not seem to be the main contributions. What are the main contributions of your paper actually?

---

> > > ### Author Response · Authors · 2023-08-22
> > >
> > > Dear Reviewer qebj,
> > >
> > > You raise an excellent point! Our prior work introducing FedRule was based on a much smaller subset of the Wyze Rule dataset, comprising only 200,000 rules. In contrast, this paper introduces the complete dataset in its full scale - over 1 million rules from 300,000 users. The dramatically larger size unlocks new research opportunities. The dataset's extensive size and heterogeneity are essential to develop and rigorously evaluate federated learning techniques, which was infeasible with the smaller subset. Furthermore, the focus here is on formally documenting the dataset and establishing benchmarks covering various learning paradigms. This aligns well with the goals of the NeurIPS Datasets and Benchmarks track. In our prior work, the focus was narrowly on presenting the FedRule algorithm and minimal discussion was provided about the dataset itself or centralized algorithms.
> > >
> > > In summary, the key differences that make this paper a novel contribution are:
> > >
> > > - Releasing the full-scale dataset rather than a small subset
> > > - Thorough documentation and statistics on the dataset
> > > - Benchmarking the new dataset with learning algorithms
> > > - Formal introduction through the NeurIPS Datasets and Benchmarks track

---

> > > > ### Comment · Reviewer_qebj · 2023-08-22
> > > >
> > > > Thanks for your feedback.
> > > > I agree that the full-scale largest dataset is the main contribution. The authors need to clearly mention that a partial Wyze rule dataset is released in the previous study.
> > > >
> > > > I would like to ask you again:  It is better to discuss what we can and cannot do with and without *large-scale* Wyze rule dataset, respectively.  Can we know the performance difference of existing methods between large-scale Wyze rule and IFTTT datasets from the previous study that you mentioned?
> > > >
> > > > I summarize my concerns.
> > > > - Wyze rule dataset was partially released in the past, and this paper opens a large-scale version with its statistics.There is no comparisons between small- and large-scale Wyze rules.
> > > > - It is unclear what kind of insights we can find from benchmarks of existing methods in large-scale Wyze rules. The authors mentioned that the benchmarking of existing methods on Wyze rules and IFTTT is already done in their previous study. If large- and small-scale Wyze rules have the same characteristics, it is unclear of the necessity of large-scale Wyze rules.
> > > > - They did not use various baselines such as R-GCN, GCMC, RotatE, and HetGNN. Although the authors mentioned that [their previous works](https://dl.acm.org/doi/abs/10.1145/3576842.3582328) conducted benchmarking on various existing studies (indeed using just some methods),  it does not conclude the unnecessity of baselines in this paper.

---

> > > > > ### Author Response · Authors · 2023-08-23
> > > > > **The first public release of Wyze Rule Dataset**
> > > > >
> > > > > Thank you for summarizing the key points. We appreciate you pushing us to clearly articulate the value of releasing the full dataset. Based on your feedback, we will expand the paper to address:
> > > > >
> > > > > - This represents the first public release of the Wyze Rule dataset. Our prior work introduced algorithms using a small internal subset but did not release the data. We will add a statement clarifying that this paper marks its initial public release.
> > > > >
> > > > > - The significantly larger scale unlocks new research opportunities. In particular, it enables the development of robust models that can handle heterogeneity across a massive number of users. The problem of heterogeneity due to non-IID data was already evident even in the smaller subset of the dataset. However, benchmarking experiments on both the smaller and full datasets using the same models illustrates that this heterogeneity poses an even greater challenge as the dataset scales up. The performance gap when applying identical approaches demonstrates the increased difficulty of effectively handling heterogeneity for rule recommendation in the full dataset.
> > > > >
> > > > > - We purposefully chose these learning baselines (including the graph learning ones) to establish our benchmark. Our goal is to inspire more research into specialized algorithms for this dataset, rather than exhaustively focusing on graph learning algorithms, like the competition we are running right now. The simple baselines reflect common practices and allow fair comparisons. Nonetheless, to further strengthen the paper, we will be adding comparisons with some of the suggested graph learning methods you mentioned. This will provide additional context on how our chosen baselines compare with other established graph techniques.
> > > > >
> > > > > - Furthermore, we will highlight that a key opportunity unlocked by this dataset is developing innovative federated learning algorithms tailored to the recommendation domain. The federated recommendation provides unique challenges around privacy-preserving personalization that warrant specialized solutions beyond standard graph approaches. We aim for this dataset to spur research into novel federated learning techniques optimized for heterogeneous recommendation tasks like rule suggestion.
> > > > >
> > > > > In the camera-ready version, we will incorporate your excellent suggestions to more clearly position how the large-scale dataset can empower new research directions and the rationale for our focused benchmarking approach. Thank you again for the insightful feedback - it will significantly strengthen the final paper.

---

> > > > > > ### Comment · Reviewer_qebj · 2023-08-23
> > > > > >
> > > > > > Thanks for your detailed response. I understood that the Wyze rule dataset is first released in this paper.
> > > > > >
> > > > > > On the other hand, I cannot agree with the design of the benchmark and your statements.
> > > > > >
> > > > > > For example,
> > > > > >
> > > > > > >The problem of heterogeneity due to non-IID data was already evident even in the smaller subset of the dataset. However, benchmarking experiments on both the smaller and full datasets using the same models illustrates that this heterogeneity poses an even greater challenge as the dataset scales up.
> > > > > >
> > > > > > There is no evidence in this paper. Please show that larger-scale non-IID data causes more challenges empirically.
> > > > > >
> > > > > > >We purposefully chose these learning baselines (including the graph learning ones) to establish our benchmark.
> > > > > >
> > > > > > The authors use only FedRules that were originally proposed by the authors. I do not think it is purposeful to analyze the Wyze rule datasets. In other words, could you explain if the key challenges are *not* just issues for FedRules?
> > > > > >
> > > > > > I require that the authors improve the benchmark study to empirically validate that Wyze rule datasets reveal (i) new insights into existing studies and/or (ii) new challenges that cannot be evaluated by using existing datasets.
> > > > > >
> > > > > > Therefore, I will keep my score unless the authors improve the benchmark study.

---

> > > > > > > ### Author Response · Authors · 2023-08-30
> > > > > > >
> > > > > > > Dear Reviewer qebj,
> > > > > > >
> > > > > > > Thank you for the further suggestions! Again, we want to recap that our main contribution is the first open dataset for smart home rule recommendation.
> > > > > > >
> > > > > > > For the benchmark part, here are the new insights
> > > > > > >
> > > > > > > >**We need better centralized rule recommendation algorithms**
> > > > > > >
> > > > > > > Our former paper [1]  proposed GraphRule, which formulates the problem as graph link prediction instead of user-item completion (Matrix Factorization, GCMC, RGCN, and HetGNN). The algorithm structure of GCMC, RGCN, and HetGNN are nearly the same in the rule recommendation setting since there is only one user-item relationship (user, select, rule_item) while they cannot deal with multiple devices with the same type, which makes the performance of user-item completion method very limited. When recommending the top 50 rules without considering multiple devices with the same type, the hit rate of Matrix Factorization is 0.8138, GCMC is 0.8445, and GraphRule is 0.91. When recommending the top 50 rules for specific devices (not just general device type), the hit rate of Matrix Factorization and GCMC is close to 0, and GraphRule is 0.6396.
> > > > > > >
> > > > > > >
> > > > > > > Although GraphRule (uses a 2-layer GraphSage model with a 2-layer Neural Network predictor) has good performance in the small dataset. When we apply the same model to the large cross-device dataset, we observe a performance drop in the Mean Rank.
> > > > > > >
> > > > > > >
> > > > > > > |                           | Test Loss | AUC   | Mean Rank |
> > > > > > > | ------------------------- | --------- | ---- | ------------ |
> > > > > > > | GraphRule (small dataset) | 0.1997      | 0.9768 |    4.349          |
> > > > > > > | GraphRule (large dataset)      | 0.1911     | 0.9825 |       10.46       |
> > > > > > >
> > > > > > > We need better algorithms, which is also why we want to host a challenge based on the dataset. Another insight is that proposing methods to directly optimize mean rank instead of test loss can be helpful.
> > > > > > >
> > > > > > >
> > > > > > > >**Challenging on cross-device FL dataset with limited data samples and data heterogeneity**
> > > > > > >
> > > > > > > In our former work [1], FedRule (uses FedGate as the training method) can have the same performance as the centralized training in the small dataset, while FedRule (uses FedAvg as the training method) cannot converge.
> > > > > > >
> > > > > > > When running on a large dataset, FedRule (using FedAvg and FedGate) can converge but cannot have the same performance as the centralized training, which means FedAvg and FedGate cannot handle such cross-device settings.
> > > > > > >
> > > > > > >
> > > > > > >
> > > > > > > |                                      | Test Loss | AUC   | Mean Rank |
> > > > > > > | ------------------------------------ | --------- | ---- | ------------ |
> > > > > > > | FedRule (small dataset, with FedAvg) | 0.3878    |0.9521| 6.661        |
> > > > > > > | FedRule (small dataset, with FedGate)| 0.1892    |0.9804| 4.156        |
> > > > > > > | FedRule (large dataset, with FedAvg) | 0.2854      | 0.9613 |  30.92            |
> > > > > > > | FedRule (large dataset, with FedGate)              | 0.2907      | 0.9486 |  37.99     |
> > > > > > >
> > > > > > >
> > > > > > >
> > > > > > >
> > > > > > > >**Summary**
> > > > > > >
> > > > > > > The dataset can provide new insights into designing better centralized rule recommendation algorithms, new FL algorithms for cross-device settings, and the co-optimization of both parts.
> > > > > > >
> > > > > > > Your suggestions help a lot to make the paper more valuable! We will make the above discussion clearer in the paper. Future researchers can easily find their direction to work on either centralized or federated algorithms.
> > > > > > >
> > > > > > > [1] FedRule: Federated Rule Recommendation System with Graph Neural Networks

---

### Official Review · Reviewer_8wtf · 2023-07-20
**Wyze Rule review comments**

**Rating:** 8
**Confidence:** 4
**Correctness:** The evaluation method and experiment …
**Clarity:** The paper is clearly written.

**Strengths:**

1.	Abundant Real-World Data: The Wyze Rule Dataset's vast collection of over 1 million rules from 300,000 users surpasses other rule recommendation datasets, making it a valuable resource for researchers.
2.	Suitable for Federated Learning: The non-iid nature of the dataset allows researchers to explore cross-device federated learning algorithms, presenting opportunities for privacy-preserving rule recommendations.
3.	Personalized Rule Creation: The user-driven approach in creating rules sets this dataset apart from others like IFTTT, providing personalized and customized rule recommendations for individual users.


**Additional Feedback:**

nil

**Documentation:**

The document is suffificent.

**Ethics:**

nil

**Limitations:**

nil

**Opportunities For Improvement:**

1.	Absence of Device Location Information: One limitation of the dataset is the lack of device location data, which may limit research on personalized rule recommendation using location information.
2.	Comparative Visualization: While the paper mentions the dataset's advantages over others, a visual comparison, such as a figure or table, would better illustrate the unique aspects of the Wyze Rule Dataset compared to existing datasets like IFTTT.


**Relation To Prior Work:**

The proposed dataset contributes towards the field.

**Summary And Contributions:**

The authors present the Wyze Rule Dataset, a significant contribution to the rule recommendation field. This dataset boasts an extensive collection of real-world rules, exceeding existing datasets and incorporating diverse user-generated rules. The dataset's non-iid characteristics make it an ideal platform for evaluating cross-device federated learning algorithms. Additionally, the authors provide baselines testing rule recommendation algorithms in both centralized and federated settings.

---

> ### Author Response · Authors · 2023-08-21
>
> Thank you for the constructive suggestions! Here are our answers to your questions:
>
> >**Absence of Device Location Information: One limitation of the dataset is the lack of device location data, which may limit research on personalized rule recommendations using location information.**
>
> As mentioned to the reviewer YUfZ, the implicit location information of the rules is available, but we will not release nor use it for training such recommendation models due to privacy concerns. Nonetheless, this could be an important factor for such a recommendation model.
>
> >**Comparative Visualization: While the paper mentions the dataset's advantages over others, a visual comparison, such as a figure or table, would better illustrate the unique aspects of the Wyze Rule Dataset compared to existing datasets like IFTTT.**
>
> That is a great point. We do have a comparison between these two datasets in our other paper (https://dl.acm.org/doi/abs/10.1145/3576842.3582328), where we apply different algorithms on these two datasets to show the efficacy of the proposed FedRule algorithm. However, such a comparison seems to be beyond the scope of this paper, as they are pursuing a different goal than our Wyze Rule Recommendation Dataset. We will add some more details about their similarities and differences in the paper.

---

> > ### Comment · Reviewer_8wtf · 2023-08-30
> >
> > I will keep my score. Thanks

---

> > > ### Author Response · Authors · 2023-08-30
> > >
> > > Dear Reviewer 8wtf,
> > >
> > > Happy to know that we have addressed your questions! We will add more details about the Wyze Smart Home Rule Dataset compared to existing datasets like IFTTT which focuses on smartphone app interactions.

---

### Official Review · Reviewer_2wK7 · 2023-07-21
**A novel and large-scale dataset for smart home rule recommendation**

**Rating:** 6
**Confidence:** 3
**Correctness:** yes
**Clarity:** The paper is well written and organized.

**Strengths:**

- The dataset is large-scale and well organized.
- The paper provides observations in both centralized and federated scenarios.

**Additional Feedback:**

Please refer to the detailed comments

**Documentation:**

Yes

**Limitations:**

yes

**Opportunities For Improvement:**

- Except the distribution of the number of devices and rules, more analysis about the data distribution heterogeneity is suggested,  such as the distribution of the user preferences over different rules.
- The detailed settings in section 4.3 is unclear, such as the partition of the data, the number of users involved in federated training.
- More analysis in terms of federated learning is suggested. For example, how about fairness in federated training, and how about the benefits of federated training when compared with local training for each client over this benchmark?

**Relation To Prior Work:**

Yes

**Summary And Contributions:**

This paper constructs a large-scale dataset for smart home rule recommendation, which is consisted of over 300, 000 users and more than 1 million rules. Its data is well organized and the prediction task is more challenging than previous benchmark. Further, the authors conduct observation in both centralized and federated scenarios.

---

> ### Author Response · Authors · 2023-08-21
>
> Thank you for the valuable comments and suggestions! We are sure that researchers can have many interesting analyses after accessing the dataset.
>
>
> >**More analysis of the data distribution heterogeneity is suggested, such as the distribution of user preferences over different rules.**
>
> Showing the data distribution heterogeneity is a very good idea. We can make a figure about user preferences over different rules and we will include it in the main paper.
>
> >**The detailed settings in section 4.3: such as the partition of the data, and the number of users involved in federated training.**
>
> We just use the natural data partition where each user is a client with its own data. Hence the data size is not evenly distributed across clients, as it can be seen in Figure 2 in the paper. We use the data of `272664` users for training and the rest for testing, where all of the training users are active during each round of communication. We will add these details to the main body as well.
>
> >**More analysis in terms of federated learning is suggested. For example, how about fairness in federated training, and how about the benefits of federated training when compared with local training for each client over this benchmark?**
>
> Fairness in FL is an important topic and it could be an interesting extension to this work. Our main goal is to release this dataset to the public with a benchmark that can compare against. We encourage other researchers as well to explore this dataset for different aspects of training such as those mentioned. We will add more detail regarding the comparison with the local training to show the efficacy of collaborative learning in the FL setting.

---

### Official Review · Reviewer_YUfZ · 2023-07-26
**Review of Wyze Rule**

**Rating:** 7
**Confidence:** 4
**Correctness:** Yes
**Clarity:** Yes

**Strengths:**

1. The Wyze Rule dataset overcomes the challenge of the lack of broader and more representative datasets in the field of smart home rule recommendation.
2. The dataset extensively encompasses rules that establish connections between various smart devices in different client households, showcasing non-iid data heterogeneity that is highly advantageous for federated learning research.
3. The authors employed various offline metrics to assess the model performance under different settings, exploring the possibilities and opportunities associated with utilizing this dataset for such tasks.


**Additional Feedback:**

No

**Documentation:**

Yes

**Ethics:**

No ethical concern

**Limitations:**

1. The Wyze Rule dataset proposed in this paper contains data from different users. However, in real-world scenarios, the rules for smart devices set by the same user may change over time and with habits.
2. The dataset lacks correlation between device locations.
3. The evaluation lacks direct feedback obtained from user interactions, making it difficult to ascertain the explicit effectiveness or satisfaction of recommended rules.


**Opportunities For Improvement:**

The Wyze Rule Dataset proposed in this paper provides a rich and comprehensive collection of real-world rule data while ensuring user privacy protection, making it highly advantageous for exploring and advancing the field of smart home rule recommendation. Moreover, the analysis of the collected data reveals the diversity across different users, aligning with the characteristics of real-world datasets and making it suitable for research in federated learning. Additionally, the authors establish comparative and evaluation benchmarks, meticulously implementing several baselines in both centralized and federated settings to measure recommendation system performance and drive advancements in the field.
However, this paper still raises some concerns. Firstly, regarding the composition of the dataset, the Wyze Rule dataset contains data from different users. However, in real-world environments, rules set by the same user for smart devices may change over time and with habits. It would be beneficial if the authors could analyze and address this issue. Secondly, the dataset lacks the establishment of correlations between device locations, making it more challenging for recommendation systems. Lastly, the paper lacks direct feedback from user interactions, which makes it difficult to determine the satisfaction of the recommendation models trained using this dataset in real-world scenarios.


**Relation To Prior Work:**

Yes

**Summary And Contributions:**

This paper specifically proposes a large-scale Wyze Rule dataset for research in smart home rule recommendation. The dataset is sourced from Wyze Labs and covers data from a diverse user base of 300,000 individuals, collecting over 1 million rules. The Wyze Rule dataset contains abundant heterogeneous data from numerous client sources, introducing a novel evaluation opportunity for federated learning algorithms. The authors meticulously implemented several baselines in both centralized and federated settings to provide opportunities for advancing recommendation systems.

This paper provides a broad and diverse real-world dataset while ensuring user privacy protection. It reflects the specific device configurations of each user, holding significant potential for the field of smart home automation. The Wyze Rule dataset inherently exhibits non-independent and non-identically distributed characteristics, thereby offering an opportunity for research in cross-device federated learning.

---

> ### Author Response · Authors · 2023-08-21
>
> Thank you for the valuable comments and suggestions!
>
> >**Rules set by the same user for smart devices may change over time and with habits**
>
> From the historical user data in our database, most users do not change the set rule for their homes very often. The data we have used to create this dataset is mainly stable rules that have been used by the users for a long time. Nonetheless, this is a valid concern, and it might be a research direction for dynamic rule recommendation.
>
> >**The dataset lacks correlation between device locations.**
>
> We also agree that the location feature can have a great impact on the recommendation performance. We do have a proxy feature for the location, however, due to privacy reasons, we will not release those features for rules.
>
> >**The evaluation lacks direct feedback obtained from user interactions**
>
> That is a great point. We have deployed the system and are performing A/B testing. That will provide an additional form of implicit feedback to create a ground truth for our dataset. While our initial results were promising, we recognize the importance of accumulating substantial real-world evidence through rigorous comparative testing. Our ongoing A/B testing will enable us to refine our models by incorporating diverse feedback at scale, well beyond what our initial dataset allowed.

---

### Author Response · Authors · 2023-08-21
**General Message**

We would like to thank all reviewers for your careful and constructive comments. We provide responses to address their concerns. Here are two main concerns shared by reviewers:

>**Location data**

We agree that including location data can be very helpful for a recommendation system. However, due to privacy concerns, we will not use this feature in our dataset and training.

>**Algorithmic comparison**

While we have provided a comparison of some limited algorithmic solutions for the rule recommendation task on our dataset, and to some extent in our previous paper (https://dl.acm.org/doi/abs/10.1145/3576842.3582328), we want to point out that our main goal is introducing the dataset, releasing it to the public (https://huggingface.co/datasets/wyzelabs/RuleRecommendation) for enticing research studies, and encouraging researchers to develop new and innovative learning solutions (beyond what presented in the paper) to this problem that can improve upon our baselines. That is why we have started a competition based on this dataset (https://huggingface.co/spaces/competitions/wyze-rule-recommendation), which is ongoing, and hopefully can attract researchers to participate and make innovation on this task.

---

### Decision · Program_Chairs · 2023-09-22

**Decision:**

Accept (Poster)

**Comment:**

This paper introduces the Wyze Rule Dataset, a significant contribution to rule recommendation research. With over 1 million rules from 300,000 users, it surpasses similar datasets and is ideal for studying federated learning algorithms due to its non-iid properties. The authors implement various baselines and use multiple offline metrics to assess model performance.

However, the dataset could be improved by including device location information and direct user feedback. In addition, comparative visualizations would better highlight its uniqueness compared to other datasets like IFTTT. Despite these suggestions for improvement, this paper provides a valuable resource for advancing smart home rule recommendation and federated learning research.